# Some poleward movement of British native vascular plants is occurring, but the fingerprint of climate change is not evident

Quentin J. Groom

Botanical Society of the British Isles, Botany Department, The Natural History Museum, London, UK

## ABSTRACT

Recent upperward migration of plants and animals along altitudinal gradients and poleward movement of animal range boundaries have been confirmed by many studies. This phenomenon is considered to be part of the fingerprint of recent climate change on the biosphere. Here I examine whether poleward movement is occurring in the vascular plants of Great Britain. The ranges of plants were determined from detection/non-detection data in two periods, 1978 to 1994 and 1995 to 2011. From these, the centre of mass of the population was calculated and the magnitude and direction of range shifts were determined from movements of the centre of mass. A small, but significant, northward movement could be detected in plants with expanding ranges, but not among declining species. Species from warmer ranges were not more likely to be moving northward, nor was dispersal syndrome a predictor of migration success. It is concluded that simply looking at northward movement of species is not an effective way to identify the effect of climate change on plant migration and that other anthropogenic changes obscure the effect of climate.

## INTRODUCTION

Among animals, numerous studies have shown recent poleward movement and upward altitudinal shift of distribution (*Parmesan & Yohe, 2003*; *Perry et al., 2005*; *Wilson et al., 2005*; *Hickling et al., 2006*; *La Sorte & Thompson, 2007*). In the case of plants there is evidence of movement towards higher altitudinal ranges, but in contrast to animals, the evidence for poleward shifts of plants is scant (*Payette et al., 1989*; *Beckage et al., 2008*; *Holzinger et al., 2008*; *Kelly & Goulden, 2008*; *Lenoir et al., 2008*; *Leonelli et al., 2011*). *Sturm, Racine & Tape (2001)* and *Smith (1994)* are often cited, but, they concern a tiny number of species in small areas of the Arctic and Antarctic.

These poleward and altitudinal range shifts have been interpreted as the fingerprint of recent climatic warming on the biosphere (*Parmesan & Yohe, 2003*; *Root et al., 2003*; *Chen et al., 2011*). So, why is there a lack of evidence for poleward range shifts among plants? One

Corresponding author
Quentin J. Groom,
qgroom@reticule.co.uk

reason may be the disparity between rapid declines in temperature with elevation over a short distance in mountains, compared to the gradual change with latitude. *Jump, Mátyás & Peñuelas (2009)* called this the "altitude-for-latitude disparity". In elevation change the dispersal rate of plants is not such an impediment to migration. Poleward movement is often assumed to be the expected response of animals and plants whose range has warmed, though there are many reasons why this may not be the case (*VanDerWal et al., 2012*).

Another complicating factor in migration is that there are many other environmental changes that are causing range shifts among plants, such as atmospheric nitrogen deposition, grazing changes, anthropogenic dispersal, disease, general eutrophication and many forms of land use change. These other migrational pressures are not necessarily in a poleward direction. Therefore, to uncover a climatic component to latitudinal migration in plants one needs to look at many species over long distances.

Range boundaries are often used to measure rates and directions of migration (*Angert et al., 2011*). It is argued that the edges of a range will be more sensitive to change than the core range of a species. Yet, a range boundary is difficult to define unless it occurs along a physical barrier. At the edges of a species range the populations will be more diffuse and the measurements of range boundaries will be sensitive to small gains and losses, whether these are real or due to differences in recorder effort.

Another problem of interpreting range shifts in bounded areas, such as an island, is that species can only extend their range in unbounded directions. So migration can be substantially misdirected from the course it would have taken without barriers. Furthermore, if the population of a species increases in its core area it will almost inevitably extend its range boundary, even if it is no more adapted to the environment in this new range. This is the concept of sink populations which would not survive were it not for replenishment from the core (*Pulliam, 1988*). Such apparent range shifts might be expected to be greater for species with effective long distance dispersal mechanisms that can replenish sink populations. In Great Britain there is a gradient of vascular plant diversity with latitude, with fewer species in the north, therefore a general northward movement might be predicted among species with expanding ranges just because there are more species in the south.

In this study I have chosen to determine the centre of mass of a population based upon the occupancy probabilities of that species across the area. The unevenness of raw biodiversity data was corrected by using detection/non-detection data in 4 km$^2$ grid squares and by smoothing spatial differences in recording effort with spatial interpolation. The migrational change and direction are based upon the movement of the centre of mass between time periods, not the range boundary. Unlike measurements of range boundaries this measurement uses a much larger proportion of the available data and because it is generated from detection/non-detection data the recording effort is better controlled and not so sensitive to local differences at the boundary. Centre of mass changes take into account, the core distribution, leading and trailing edges of a population's migration.

The centre of mass of species with expanding ranges will move in the direction of migration, whereas the centre of mass of declining species will move away from the area of

greatest decline. As these mechanisms are different, the results are analysed separately for species with expanding and declining ranges.

The aim of this research was to examine the natural migration of plants in Great Britain. That is, the migration driven by the plants' own dispersal mechanisms and not by the deliberate dispersal by mankind through horticulture and forestry. For this reason, I have concentrated on examining the migration of native species. I aim to establish whether there is a poleward migration of plants within Great Britain and whether the movements of plants can be explained by climate change and dispersal syndrome.

## MATERIALS & METHODS

Species occupancy maps were created for all but the rarest species in Great Britain as has previously been described in *Groom (2013)*. In summary, records were used from the Distributions Database of the Botanical Society of the British Isles (BSBI). All dated records from 1978 to 2011 were used in this study from all parts of Great Britain, except for the islands of the Outer Hebrides, Shetland, the Channel Islands and the Isle of Man. A snapshot of the database was taken in November 2011 (*Botanical Society of the British Isles, 2011*). Any records of subspecies and variety were amalgamated with those of the species to which they belong. Microspecies in the genera *Taraxacum*, *Hieracium*, *Euphrasia* and *Rubus* were combined with records of the aggregate species, as were some other taxa such as *Arctium, Cotoneaster*, and *Rosa canina agg.* Taxonomy follows *Stace (2010)*. The grid system used is that of the Ordnance Survey of the United Kingdom.

Estimation of species ranges was done by selecting well-surveyed grid cells ($4 \text{ km}^2$) from a pool of all records and generating detection/non-detection data from these. The spatial distribution of occupancy is modelled using variograms and these models are used to interpolate the occupancy probability across the whole area in a process called kriging. To avoid kriging over landscapes with different spatial structures Great Britain was separated into four partitions, Scotland, Wales, northern England and southern England. Northern England was defined as is traditional for the BSBI as vice counties including, and northwards from, South Lincolnshire, Leicestershire, Derbyshire and Cheshire (for biological recording purposes Great Britain is divided into 113 vice counties with fixed borders). For convenience these regions and countries will be referred to as the partitions. Partitioning the area is not entirely necessary, but does allow statistical comparison between partitions and species.

Species without at least 50 occupancies in a partition were not considered in the analysis. Rare species tend to have the greatest biases in their recording and in the conservation efforts used to preserve them in their localities.

Interpolated maps were created for two time periods 1978 to 1994 and 1995 to 2011. These periods were chosen because they are of equal length and both periods contain a national sample survey of $4 \text{ km}^2$ grid squares. Selection of sample squares, creation of detection/non-detection data and the resulting estimates of recording effort, average occupancy and occupancy change are all described in detail in *Groom (2013)*. Well-surveyed grid cells are defined as those that have had at least two days of surveying

conducted in them and a minimum threshold of species recorded in each of those surveys. Details of these thresholds and discussion of the differences in recording effort between the different time periods is given in *Groom (2013)*. Essentially, the process of kriging balances the spatial differences in recording and the selection criteria of grid cells balancing temporal differences. Furthermore, although the selection threshold is important, the method is insensitive to its precise value.

Centre of mass was calculated using the following formula, where $x'$ and $y'$ are the coordinates of the centre of mass; $x_i$ and $y_i$ are the coordinates of each grid square and $o_i$ is the predicted occupancy probability of grid square $i$.

$$x' = \frac{\sum_{i=1}^{n} o_i x_i}{\sum_{i=1}^{n} x_i} \qquad y' = \frac{\sum_{i=1}^{n} o_i y_i}{\sum_{i=1}^{n} y_i}.$$

A custom made Perl (version 5.12.4) script was used to calculate the coordinates of the centre of mass from the occupancy probabilities calculated from kriging. The distance moved by the centre of mass between time periods was calculated using Euclidean geometry. Namely, Pythagoras' theorem was used to calculate the magnitude of movement from the coordinates and trigonometric functions were used to calculate the direction.

The mean July & January temperatures of the ranges of each species were taken from *Hill, Preston & Roy (2004)*. Dispersal syndromes were taken from *Klotz, Kühn & Durka (2002)* and *Fitter & Peat (1994)*.

Statistical analysis and kriging was conducted using R, version 2.8.1. Variogram creation, fitting and kriging were conducted using the package GSTAT, version 1.0–10 (*Pebesma & Wesseling, 1998*). Data manipulations and reformatting were conducted in MS-Excel. Circular means and circular bootstrap confidence intervals where calculated using the R package "Circular" Version 0.4-3 (*Agostinelli & Lund, 2011*).

## RESULTS

Among species with increasing ranges northern and southern England have distinctly bimodal directions of migration. Scotland has a unimodal, northern direction and Wales has a more scattered dispersal of directions (Fig. 1). The circular average of all the directions from all four partitions is 356.8° ($n = 1243$), that is, northward. Assuming all the directions conform to a von Mises distribution the bootstrap 95% confidence interval is from 333.7° to 20.6° (the von Mises distribution for circular data is equivalent of the normal distribution for linear data). Given that some of the partitions have bimodal distributions this is a weak assumption, but it does give an indication of an overall northward movement of vascular plants in Britain. Indeed, even without this assumption, if the direction of movement is treated as a coin toss between north and south the probability of a species moving north for species with increasing occupancy is significant ($p < 0.05$) (Table 1), though the northward movement is not significant in Wales and southern England when taken alone.

The directions of movement of the centre of mass of native species with decreasing occupancy are also bimodal or perhaps multimodal (Fig. 2). Their average direction of

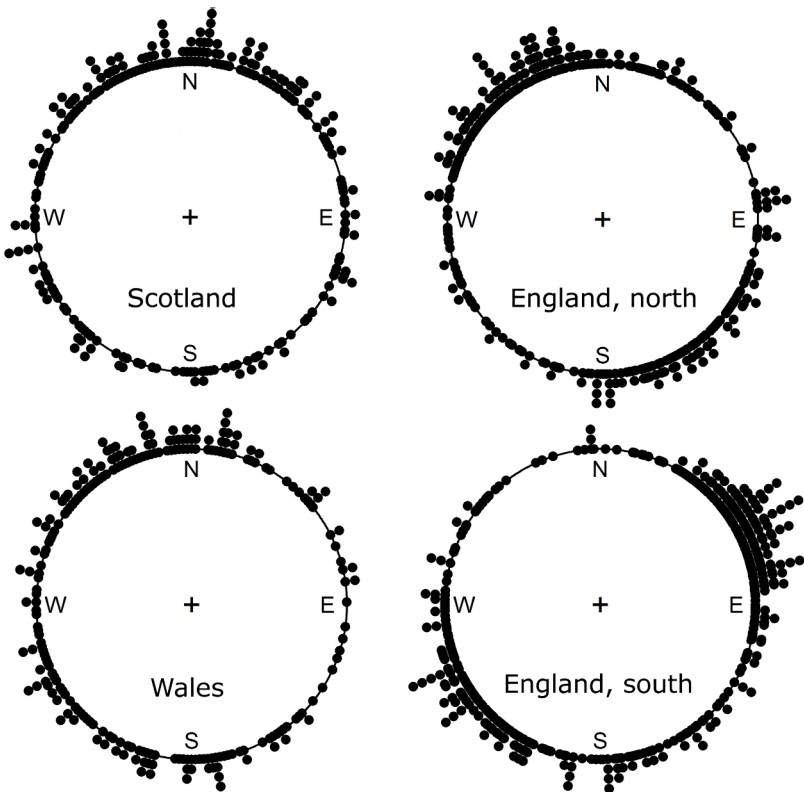

**Figure 1 A circular histogram of the directions of movement of the centre of mass for those native species with increasing occupancy rates.** Each dot represents the direction of migration for one species. All distributions are significantly ($p < 0.05$) different from random using a Kuiper test. Directions of all species are available in Table S1.

**Table 1 The proportion of species moving northwards in each of the four partitions for plants with increasing occupancy.** The overall average is calculated as if the four separate partitions were replicates of the same experiment ($n = 4$).

| Partition | Number of species moving north | Total number of species | Proportion moving northward and 95% confidence interval |
|---|---|---|---|
| Scotland | 163 | 251 | 0.65 (0.59–0.71) |
| Northern England | 189 | 331 | 0.57 (0.52–0.62) |
| Wales | 133 | 238 | 0.55 (0.50–0.62) |
| Southern England | 216 | 423 | 0.51 (0.46–0.56) |
| All | | | 0.56 (0.51–0.62) |

movement was roughly similar to increasing species in the case of northern England and Wales, but approximately opposite for Scotland and southern England. Overall, the circular average direction was 264.5°, that is westerly, bootstrap 95% confidence interval is from 319.7° to 208.5°, under the von Mises distribution assumption. Treating the migration as a coin toss between north and south there is no significant northerly movement of declining species overall (Table 2). However, the southerly trend for Scotland

**Peer**J

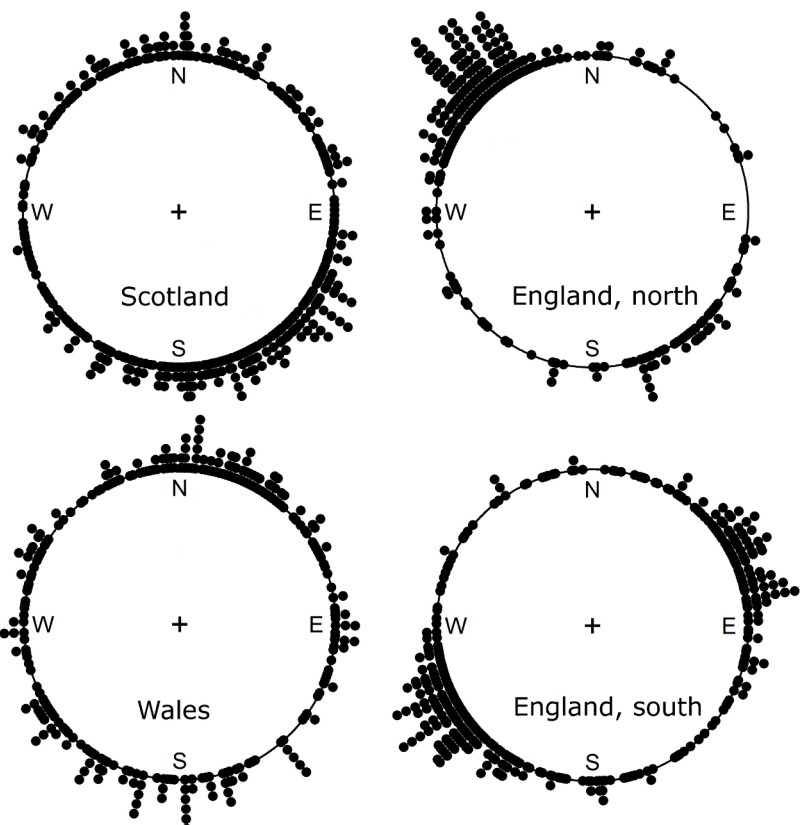

**Figure 2 The direction of movement of the centre of mass for those native species with decreasing occupancy rates.** Each dot represents the direction of migration for one species. All distributions are significantly ($p < 0.05$) different from random using a Kuiper test. Directions of all species are available in Table S1.

**Table 2 The proportion of species moving northwards in each of the four partitions for plants with decreasing occupancy.** The overall average is calculated as if the four separate partitions were replicates of the same experiment ($n = 4$).

| Partition | Number of species moving north | Total number of species | Proportion moving northward and 95% confidence interval |
|---|---|---|---|
| Scotland | 147 | 382 | 0.38 (0.33–0.43) |
| Northern England | 184 | 263 | 0.70 (0.63–0.76) |
| Wales | 156 | 280 | 0.56 (0.50–0.62) |
| Southern England | 143 | 356 | 0.40 (0.35–0.45) |
| All | | | 0.53 (0.35–0.64) |

and southern England and the northerly trend for northern England are significant ($p < 0.05$).

If climate warming was a strong driver of migration in Great Britain we might expect increasing species from warmer areas to be moving northward, whereas increasing species from colder areas would be migrating in directions unrelated to the climate. The species

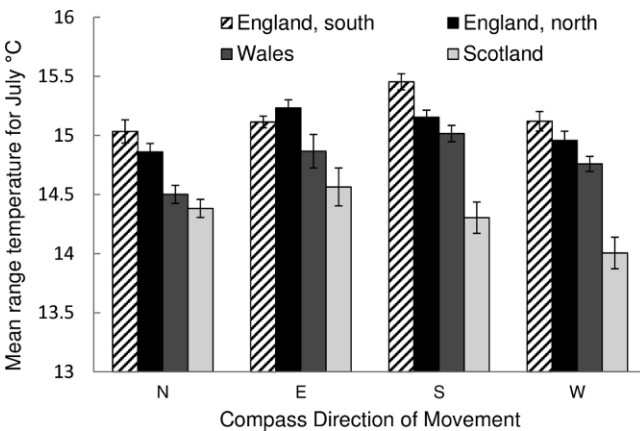

**Figure 3 The mean July temperature of the ranges of species for the four different area partitions of the study.** Species which had increased occupancy over the period of this study are split by the direction of movement of their centre of mass, north, south, east or west. Error bars are two standard errors of the mean. The number of species contributing to each value are as follows, Scotland N-112 S-44 E-43 W-51, England, north N-117 S-82 E-61 W-71, Wales N-86 S-64 E-24 W-64, England, south N-48 S-98 E-174 W-103.

were separated into four groups based upon the compass direction of their movement, north, south, east and west. The averages of the mean July temperatures of each group were compared. For all partitions, the average range temperature for species moving north was either similar or lower than for other compass directions; whether or not the species are increasing or declining (Fig. 3 shows the results for species with increasing occupancy). Similar negative results were found for mean January temperatures and declining species. No obvious pattern emerges; plants from warmer ranges are not more likely to be moving northward.

Nevertheless, mean July temperatures of the species ranges are positively correlated with the relative occupancy change of all species, whether increasing or decreasing, except for in Wales where there is no correlation (Scotland $R^2 = 0.14$, $n = 661$; northern England $R^2 = 0.54$, $n = 627$; Wales $R^2 = 0$, $n = 556$, southern England $R^2 = 0.14$, $n = 838$). So there is an indication that species from warmer ranges are increasing and species from colder ranges are declining, though this requires further investigation as there are many co-correlates that could lead to this result.

No significant differences were found when comparing the magnitude of the movement of the centre of mass with dispersal syndrome (Fig. 4). This was also examined in another manner. As small populations can move their centre of mass relatively easily compared to widespread, common species, a measurement analogous to linear momentum might be a more useful metric of migration i.e., velocity multiplied by mass. We can look at migration as the product of the magnitude (km) and the absolute change in occupancy. In this case, time is constant so magnitude is used as a proxy for velocity. Nevertheless, there was still no significant difference in the momentum of migration and the dispersal syndrome (results not shown).

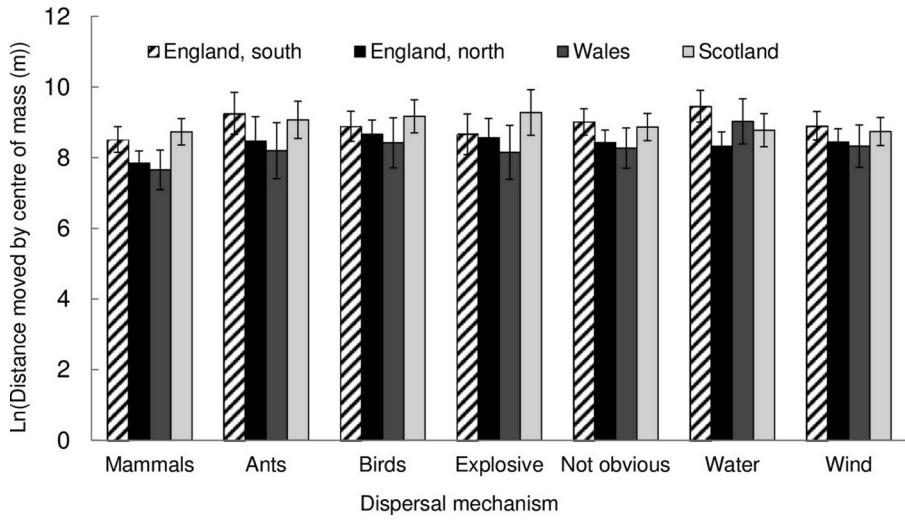

**Figure 4 The natural log of the distance moved by the centre of mass for different dispersal mechanisms of species with increasing occupancy. Error bars are two standard errors of the mean.** The number of species in each group were for England, south - Mammals 15, Ants 9, Birds 41, Explosive 10, Not obvious 249, Water 29, Wind 69. England, north - Mammals 12, Ants 4, Birds 33, Explosive 8, Not obvious 178, Water 32, Wind 63. Wales - Mammals 6, Ants 6, Birds 10, Explosive 7, Not obvious 143, Water 20, Wind 43. Scotland - Mammals 8, Ants 8, Birds 14, Explosive 4, Not obvious 141, Water 13, Wind 58.

Given that northern migration of the centre of mass cannot be easily explained by climate and that there is no obvious influence of dispersal syndrome on the magnitude of movement it is informative to look at examples of species with large movements in their population's centre of mass (Table 3).

The most obvious group among these species are the halophytes e.g., *Atriplex littoralis, Beta vulgaris, Cochlearia danica, Puccinellia distans* and *Spergularia marina*. Yet, there are no common directions in the movement of these plants. Other common features are far less clear. Orchids are quite well represented e.g., *Dactylorhiza maculata, D. praetermissa* and *Goodyera repens* as are other wind dispersed plants such as *Acer campestre, Lactuca virosa, Phragmites australis, Polystichum setiferum, Populus nigra, Sonchus asper* and *Typha latifolia*. Yet there are several other dispersal strategies represented, including animal dispersed species (*Bryonia dioica, Rosa caesia, Rubus caesius* and *Solanum dulcamara*) and water dispersed plants (*Oenanthe crocata* and *Comarum palustre*).

## DISCUSSION

The centre of mass in bounded ranges will tend to move parallel with the long axis of the area. For example, the south of England is very roughly a right-angle triangle with the acute angle in the west. If a species has its core range in the west, but for climatic reasons is able to grow further north, its centre of mass will move north-eastward as it occupies more northerly territory, because it is blocked from moving directly north by the sea and the boundary with northern England. This explains why the majority of centre of mass

**Table 3 Examples of native species with large changes in their centre of mass and their absolute occupancy.** Species with large changes in their centre of mass were selected by having the highest product of their absolute change in occupancy probability and the distance that their centre of mass moved. Directions are from zero at grid north. Distance is the distance moved by the centre of mass. Mean occupancy probability and absolute change are taken from *Groom (2013)*. Details of all species are available in Table S1.

| | Direction (degrees) | Distance (km) | Mean occupancy probability per 4 km$^2$ | Absolute change |
|---|---|---|---|---|
| **England, south** | | | | |
| *Cochlearia danica* | 40 | 72.4 | 0.153 | 0.156 |
| *Oenanthe crocata* | 230 | 59.9 | 0.394 | 0.073 |
| *Lactuca virosa* | 266 | 44.9 | 0.096 | 0.078 |
| *Polystichum setiferum* | 235 | 46.1 | 0.336 | 0.070 |
| *Puccinellia distans* | 273 | 51.5 | 0.083 | 0.054 |
| *Beta vulgaris* | 35 | 46.9 | 0.129 | 0.057 |
| *Spergularia marina* | 18 | 32.8 | 0.079 | 0.077 |
| *Rubus caesius* | 70 | 42.8 | 0.295 | 0.052 |
| *Hypericum androsaemum* | 219 | 17.0 | 0.234 | 0.123 |
| *Atriplex littoralis* | 342 | 30.2 | 0.063 | 0.066 |
| **England, north** | | | | |
| *Lactuca virosa* | 324 | 90.1 | 0.103 | 0.064 |
| *Acer campestre* | 322 | 36.1 | 0.508 | 0.123 |
| *Bryonia dioica* | 144 | 43.9 | 0.159 | 0.090 |
| *Populus nigra* | 170 | 26.7 | 0.165 | 0.121 |
| *Rosa arvensis* | 181 | 26.9 | 0.265 | 0.108 |
| *Apium nodiflorum* | 163 | 19.7 | 0.280 | 0.146 |
| *Carex otrubae* | 145 | 19.4 | 0.245 | 0.133 |
| *Spergularia marina* | 96 | 15.1 | 0.194 | 0.164 |
| *Solanum dulcamara* | 177 | 13.6 | 0.467 | 0.170 |
| *Phragmites australis* | 142 | 11.7 | 0.348 | 0.192 |
| **Wales** | | | | |
| *Dactylorhiza praetermissa* | 78 | 45.0 | 0.169 | 0.097 |
| *Comarum palustre* | 28 | 29.5 | 0.231 | 0.092 |
| *Baldellia ranunculoides* | 196 | 71.6 | 0.036 | 0.030 |
| *Dactylorhiza maculata* | 247 | 10.8 | 0.243 | 0.177 |
| *Carex muricata* | 107 | 19.8 | 0.177 | 0.093 |
| *Ornithopus perpusillus* | 171 | 13.5 | 0.174 | 0.122 |
| *Vulpia bromoides* | 6 | 20.9 | 0.251 | 0.071 |
| *Carex otrubae* | 196 | 12.0 | 0.252 | 0.122 |
| *Erica cinerea* | 340 | 12.4 | 0.303 | 0.117 |
| *Fumaria bastardii* | 150 | 36.5 | 0.111 | 0.037 |
| **Scotland** | | | | |
| *Goodyera repens* | 6 | 80.0 | 0.040 | 0.035 |
| *Rumex longifolius* | 31 | 32.4 | 0.146 | 0.056 |
| *Anthriscus sylvestris* | 334 | 33.9 | 0.416 | 0.051 |
| *Rosa caesia* | 336 | 14.9 | 0.192 | 0.112 |

Table 3 (*continued*)

| | Direction (degrees) | Distance (km) | Mean occupancy probability per 4 km$^2$ | Absolute change |
|---|---|---|---|---|
| *Sonchus asper* | 345 | 10.5 | 0.451 | 0.122 |
| *Spergularia marina* | 60 | 12.0 | 0.144 | 0.097 |
| *Dactylorhiza maculata* | 321 | 7.4 | 0.436 | 0.150 |
| *Lycopus europaeus* | 20 | 31.2 | 0.084 | 0.034 |
| *Typha latifolia* | 353 | 32.3 | 0.092 | 0.030 |
| *Pyrola media* | 13 | 56.7 | 0.032 | 0.017 |

movements are on a north-easterly to south-westerly axis for southern England (Fig. 1). The other, more rectangular, partitions show a north-south axis.

Recent poleward migration has been repeatedly claimed for animal species in a number of countries including Great Britain. Yet, in this study it can only be confirmed for plants with expanding ranges and then only weakly. Furthermore, recent reanalysis of avian range margin shifts has shown that changes largely disappear when recording effort is correctly accounted for *Kujala et al. (2013)*.

Among the plants that are moving northward there is no evidence that these are from warmer climates. Perhaps it is unrealistic to already expect a northward movement of plant species due to climate change. The mean annual temperature for the UK has only risen about 0.25°C over the period of this study (*Met Office, 2010*), which is perhaps too small to have a significant impact. Also, a lag is to be expected in the reaction of plants to climate change. At the leading edge of migration a lag will occur because of the limitations of natural dispersal (*Menéndez, González Megías & Hill, 2006*). While at the trailing edge, a lag will occur because of the persistence of perennial species in otherwise unfavourable climates (*Jump, Mátyás & Peñuelas, 2009*).

Furthermore, the effects of climate change are more complicated than a simple northward shift of range. For example, there are examples of species migrating in the opposite direction to that originally predicted from temperature changes (*Hilbish et al., 2010*; *Lenoir, Gegout & Guisan, 2010*; *Crimmins et al., 2011*). Modelling of the climatic niche of Australian birds showed that a complex interaction of temperature and precipitation change predicted a wide range of migration directions. If only poleward movement was considered the impact of climate change was seriously underestimated (*VanDerWal et al., 2012*). In Great Britain, migration could occur towards the more temperate coasts to avoid greater temperature extremes in the centre of the country. In which case, some species could move southward in response to temperature change.

Great Britain, like many temperate countries, has many latitudinal gradients. These gradients are a consequence of geology and geography and only partially related to the climatic gradient. For example, human population and soil fertility decrease towards the north, while altitude increases. The greater population in the south means more disturbance in the south, greater introduction of alien species and more transportation. The lower human population in the north, higher elevation and infertile soils means more extensive farming methods and forestry. These non-climatic gradients might act as barriers

to the northward migration of species. Also, Britain has a mild oceanic climate, which could soften the impact of climate change.

Perhaps it is counterintuitive, but different dispersal strategies made no difference to the migration of plants in Great Britain during this period. Species traits were also a poor predictor of range shifts in North American Passeriformes and British Odonata (*Angert et al., 2011*). Even though many wind-dispersed species were among the top migrating species, halophytes, with no obvious morphological dispersal strategy, moved just as rapidly. Halophytes have the advantage of an uninterrupted habit, free from competitors as they spread along roads where salt is strewn in the winter (*Scott & Davison, 1982*). Clearly, the dispersal strategy is not always the rate limiting factor in migration and habitat availability is important.

The fingerprint of climate change is not yet obvious on the migration of plants in Great Britain. Even though climate change is affecting British plants in other ways, such as changes in phenology (*Sparks, Jeffree & Jeffree, 2000*). This is not to say that migration due to climate change will not, or has not occurred, however, its traces in Great Britain are obscured by other manmade changes to the environment and will require more sensitive analyses to uncover.

## ACKNOWLEDGEMENTS

Greatest thanks go to those numerous volunteers who collect and share data on wild plants in the British Isles. I also appreciate encouragement and guidance received from Sandrine Godefroid and the helpful comments of Ivan Hoste during the preparation of this paper.

### Funding

There was no external funding for this work.

### Competing Interests

The author declares no competing interests.

### Author Contributions

- Quentin J. Groom conceived and designed the experiments, performed the experiments, analyzed the data, contributed reagents/materials/analysis tools, wrote the paper, made other contributions.

### Supplemental Information

Supplemental information for this article can be found online at http://dx.doi.org/10.7717/peerj.77.

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
