# Peer review of "Some poleward movement of British native vascular plants is occurring, but the fingerprint of climate change is not evident"

_PeerJ, doi:10.7717/peerj.77_

## Round 0.1 · original submission · Minor Revisions

Beyond those comments of the reviewers, in Figs 1 and 2 there are some clearly bimodal plots, and when that occurs, then I don't think that circular means are useful. Finally apologies for the delay, I have had problems finding willing reviewers.

·

Basic reporting

‘Occupancy’, ‘detection/non-detection’ etc. are not actually defined. Some later parts, such as the calculation of momentum, are not easy to follow because of this.
The presentation comes over as a little parochial: for an international audience it would be worth explaining terms such as ‘vice-county’.
‘That is’ in P6 L20 is not a very clear or grammatical way of starting a sentence.
Circular averages that are the marginal resultant of largely counterbalancing values are not very convincing intuitively but I suppose this is the nature of circular geometry.
In Figs 3 & 4 it would be helpful to give the number of species contributing to each category.
In Tables 1 & 2 the overall average is the mean of the four partitions. Presumably there are many species in more than one partition and so there would be multiple accounting for these? These two tables could be conflated.

Experimental design

The details of the methodology are presented in another paper, in press, and not available to the reviewer. So all this must be taken on trust. A reference should be given for the BSBI distribution database.

Validity of the findings

I am happy with these.

Additional comments

There are many reasons why British plants would not show poleward (or even other) shifts in distribution in response to climatic change but these are all well discussed or accounted for in the paper. It was well worth investigating, given such a good database (and so many species) to work with, even though it was a short period in plant dispersal terms, in which there had been a tiny amount of average warming. It would be interesting to repeat it in another 20 years.

Reviewer 2 ·

Basic reporting

The paper is well written, with citation of appropriate literature.

Methods: reference to Groom (2013), an unpublished paper, could not be accessed.

Experimental design

Data appears only to be available from 1978 onwards, and probably recording has increased markedly over this time period? Comparison of the two adjacent 16 year periods 1978-1994 and 1995- 2011 gives weak findings.

The partitioning of England into N & S appears rather arbitrary? Is it supported by the natural distribution of the species recorded? For example, Goodyera repens, one of the selected northern species, has a southern range limit across Northumbria and Cumbria.

Was the data examined for significant collector bias? Any evidence of repeat collections or specific targeted areas, not previously surveyed?

Validity of the findings

The work is interesting, but the results should not be over-interpreted:

- other major factors may outweigh the apparent temperature effects e.g. habitat loss/land use/conservation effort/human activity/reintroductions? Some British species are thought to have suffered range contraction linked to habitat loss/ change of land use, and in an opposite direction to that expected by considering climate change.

- with only a rise of 0.25°C in annual temperature over this 32-year period, and with time lags in plant migration it’s not unexpected that signals of migration are weak. This could be clarified in the abstract.

---

## Round 0.2 · accepted · Accept

None - many thanks for your submission